# Estimation of Various Walking Intensities Based on Wearable Plantar Pressure Sensors Using Artificial Neural Networks

**DOI:** 10.3390/s21196513

**Published:** 2021-09-29

**Authors:** Hsing-Chung Chen, Ben-Yi Liau, Chih-Yang Lin, Veit Babak Hamun Akbari, Chi-Wen Lung, Yih-Kuen Jan

**Affiliations:** 1Department of Computer Science and Information Engineering, Asia University, Taichung 41354, Taiwan or shin8409@ms6.hinet.net (H.-C.C.); sunardi@umy.ac.id (S.); 2Department of Medical Research, China Medical University Hospital, China Medical University, Taichung 404333, Taiwan; 3Department of Mechanical Engineering, Universitas Muhammadiyah Yogyakarta, Yogyakarta 55183, Indonesia; 4Department of Biomedical Engineering, Hungkuang University, Taichung 433304, Taiwan; byliau@hk.edu.tw; 5Department of Electrical Engineering, Yuan Ze University, Chungli 32003, Taiwan; andrewlin@saturn.yzu.edu.tw; 6Department of Creative Product Design, Asia University, Taichung 41354, Taiwan; 109711569@live.asia.edu.tw; 7Rehabilitation Engineering Lab, University of Illinois at Urbana-Champaign, Champaign, IL 61820, USA; 8Kinesiology and Community Health, University of Illinois at Urbana-Champaign, Champaign, IL 61820, USA; 9Computational Science and Engineering, University of Illinois at Urbana-Champaign, Champaign, IL 61820, USA

**Keywords:** artificial neural network, automatic classification, plantar region pressure image, walking speed, walking duration

## Abstract

Walking has been demonstrated to improve health in people with diabetes and peripheral arterial disease. However, continuous walking can produce repeated stress on the plantar foot and cause a high risk of foot ulcers. In addition, a higher walking intensity (i.e., including different speeds and durations) will increase the risk. Therefore, quantifying the walking intensity is essential for rehabilitation interventions to indicate suitable walking exercise. This study proposed a machine learning model to classify the walking speed and duration using plantar region pressure images. A wearable plantar pressure measurement system was used to measure plantar pressures during walking. An Artificial Neural Network (ANN) was adopted to develop a model for walking intensity classification using different plantar region pressure images, including the first toe (T1), the first metatarsal head (M1), the second metatarsal head (M2), and the heel (HL). The classification consisted of three walking speeds (i.e., slow at 0.8 m/s, moderate at 1.6 m/s, and fast at 2.4 m/s) and two walking durations (i.e., 10 min and 20 min). Of the 12 participants, 10 participants (720 images) were randomly selected to train the classification model, and 2 participants (144 images) were utilized to evaluate the model performance. Experimental evaluation indicated that the ANN model effectively classified different walking speeds and durations based on the plantar region pressure images. Each plantar region pressure image (i.e., T1, M1, M2, and HL) generates different accuracies of the classification model. Higher performance was achieved when classifying walking speeds (0.8 m/s, 1.6 m/s, and 2.4 m/s) and 10 min walking duration in the T1 region, evidenced by an F1-score of 0.94. The dataset T1 could be an essential variable in machine learning to classify the walking intensity at different speeds and durations.

## 1. Introduction

Walking has been universally recommended as a rehabilitation strategy to improve physical and psychological health in people with Parkinson’s disease [1], diabetes mellitus (DM), and peripheral arterial disease [2,3]. Regarding the characterization of the walking volume (i.e., different speeds and durations), the American Physical Activity Guidelines recommend that people should engage in moderate-intensity physical activity for at least 150 min/week or vigorous-intensity physical activity for at least 75 min/week [2]. Walking is an effective rehabilitation intervention to improve the health of DM patients [4]. However, moderate and vigorous physical activity, such as brisk walking, will increase the load on plantar soft tissues for causing high peak plantar pressure (PPP). An increase in walking duration will increase the repetitive load on plantar soft tissues, which results in increased stiffness and PPP [5,6,7]. Increased stiffness of plantar soft tissue has been considered as a risk factor of foot ulcers in DM patients [8]. Appropriate walking intensity has been shown to reduce plantar soft tissue stiffness [7] and decrease PPP [9]. Measurement of walking intensity is a major challenge fundamentally because the development of foot ulcers is influenced by repetitive loads on the plantar soft tissues.

Researchers have attempted to address this situation through analysis of plantar pressure patterns to study the development of foot ulcers [10,11]. The plantar pressure patterns are essential information to detect the development of foot ulcers, which have enabled to reach state-of-the-art predictive performance in classification tasks [12]. Furthermore, plantar pressure values, such as PPP, vary during gait in different areas of the plantar foot [13]. Research studies analyzed PPP values and demonstrated that four plantar regions are at a higher risk of foot ulcers, namely first toe (T1), first metatarsal head (M1), second metatarsal head (M2), and heel (HL) [14,15]. Fundamentally, the tissue thickness of the T1 is thinner than M1, M2, and HL [9]. This argument led this study to introduce the hypothesis: the T1 region may play a significant role in the development of foot ulcers. Hence, analyzing the four plantar regions may provide more detailed patterns of the foot plantar thus could be used to assess the appropriate intensity of exercise for patients at risk for foot ulcers.

Several approaches have been implemented to analyze the pattern inside the images, one of the most superior is the application of machine learning [16,17]. Machine learning methods such as artificial neural networks (ANN), can be used to classify plantar pressure patterns result in a faster and more accurate diagnostic process to aid in treatment, early detection, and prevention strategies [18]. The ANN is a machine learning method that replicates how experienced experts solve problems. The ANN learns through experience by establishing patterns and relationships from datasets. Recent studies have applied the ANN model for plantar pressure image classification and estimation. Rupérez et al. showed that the ANN model achieved satisfactory accuracy in estimating the maximum pressure on the plantar surface exerted by a plantar pressure image for three distinct phases of the gait cycle [19]. The ANN model is an automated solution that can classify different walking speeds from different patterns of plantar pressure images [20]. However, this model is only tested for the classification of walking speed. On the other hand, the duration of walking (i.e., repetitive stress on the soft tissues of the plantar foot) may increase the risk of foot ulcers [2]. According to Bowling et al., reduced soft tissue thickness can contribute to greater stiffness (i.e., increased risk of foot ulcers), especially with repetitive loads at longer walking duration [21]. Therefore, the walking duration is important for DM patients who are at risk of foot ulcers. The ANN could provide an appropriate architecture to infer features, from plantar pressure images, that accurately classify the exercise volume. However, regarding the latest studies that utilize the ANN model, there is no research that has considered walking speed and walking duration for plantar foot pressure classification.

Based on the above studies, walking exercise is an excellent intervention to improve the health of people with DM, but the appropriate walking intensity is critical. Therefore, it is necessary to classify walking speed and walking duration to find the proper walking intensity. In this work, the credential of ANN is implemented as a potential classifier of walking speed and walking duration in three different ways: (1) Demonstrate that ANN can accurately predict walking speed and walking duration using plantar region pressure images and compare predictions with preliminary approaches; (2) establish the region analysis of predictions using a dataset of four different regions; (3) assess the proposed scheme’s reliability using cross-validation of F1-scores. The results of this study provide a foundation for understanding the pattern of plantar pressure during various walking activities for detecting appropriate exercise volume in patients at risk for foot ulcers.

## 2. Materials and Methods

This section describes the plantar pressure recording, the labeling process, and the design of the classification model. The dataset of plantar pressure images was recorded from the participants who walked at different speeds and durations. This dataset is a set of plantar pressure images that represent pressure acting on the plantar foot. As the classification model, an ANN model was used to classify the dataset of plantar pressure images. The ANN model has gained popularity due to its ability to extract unique representations from image data [22]. The ANN model could be utilized to extract image properties (i.e., statistical stationarity and pixel-dependent locality) for image classification [23].

Modifying the ANN model, such as a multi-layer perceptron with sigmoid activation rules and two hidden layers, provides better performance capabilities [24]. The more hidden layers are used, the better the ANN model learns more complex patterns in a dataset. However, the large number of hidden layers did not guarantee an increase in performance. On the contrary, it wasted data processing resources [25]. Therefore, this study applied two hidden layers to extract data patterns efficiently while keeping resources to a minimum. Finally, this study presented an ANN model to improve performance for classification at various walking speeds and durations.

### 2.1. Participants

Twelve healthy participants between the ages of 18 and 45 years who can walk independently without using an assistive device were recruited from the University of Illinois at Urbana-Champaign and the nearby community. Participants who were not eligible included those having active foot ulcers, diabetes mellitus, pain in any lower extremity joint, obvious foot deformities (flat foot or high arch) [26], a history of foot amputation, or any other lower extremity surgery. The study received the approval of the Institutional Review Board of the University of Illinois at Urbana-Champaign. All participants signed an informed consent form before screening and examination procedures.

### 2.2. Procedures

The experimental procedures were based on our previous study [9,27]. Meanwhile, this study examined three walking speeds (slow at 0.8 m/s, moderate at 1.6 m/s, and fast at 2.4 m/s) and two walking durations (10 and 20 min). Three speeds were selected to simulate three common speeds in adults [28], and two durations were selected based on average walking time and physical activity [29]. All participants were asked to walk on a treadmill at speeds of 0.8 m/s (first week), 1.6 m/s (second week), and 2.4 m/s (third week) in a room with a temperature of 24 ± 2 °C. Each week, participants were assigned randomly to either 10 or 20 min of walking. Then, the participant rested for at least 20 min between two walking trials to avoid carryover effects and muscle fatigue. The wearable plantar pressure measurement system, F-Scan^®^ (Tekscan Inc., South Boston, MA, USA), was utilized to record plantar pressure images during continuous walking [8,9]. The sampling frequency was set to 300 Hz. The F-Scan^®^ in-shoe system provides dynamic pressure information to foot function and gait analysis. The F-scan^®^ in-shoe sensor has 960 sensing elements that are constructed using patented thin-film sensors. Each sensing element measures biological tissue with an area of 5.08 mm × 5.08 mm. The participants were provided with a suitable pair of shoes and socks (Altrex, Teaneck, NJ, USA) equipped with F-scan^®^ in-shoe sensors located between the socks and insoles. Before the walking experiment, participants were instructed to walk for 3 to 5 min to adapt the shoes.

### 2.3. Labeling

The labeling stage is critical for the ANN model as the system will learn classification tasks from an extensive collection of images that have been labeled differently for each category [30]. The ANN model that trains labeled data can improve performance in solving classification problems [31]. Data labeling is often performed to reference the ground truth. Ground truth labeling is a vital step in classification tasks as it provides a foundation for the performance evaluation of the proposed model [32]. This study used the labeling process to categorize the plantar pressure image data into six categories: a slow walking speed of 0.8 m/s, a moderate walking speed of 1.6 m/s, and a fast walking speed of 2.4 m/s with two walking durations (10 and 20 min). The examples of plantar pressure distribution images of the individual participant at different walking speeds and walking durations are presented in Figure 1.

### 2.4. Pre-Processing of Training Model

Image segmentation into regions of interest (ROI) aims to make an image more meaningful and easier to analyze and interpret [32]. Image cropping could also be implemented as an image resizing operation and ROI extraction. ROI on the plantar foot is divided into four regions consisting of T1, M1, M2, and HL based on the risk for foot ulcers [14,15]. A crucial role in image processing is the ROI analysis method because it identifies features and patterns in the image [33]. Analyzing the plantar pressure regions characterized by the ROI can improve the performance of the classification model than examining pressure on the entire foot [30]. The plantar pressure image of each region was resized from 5 × 5 pixels to 100 × 100 pixels using MATLAB R2019b (MathWorks, Inc., Natick, MA, USA). Resizing the image of each region would increase the performance of the classification model [34]. Resized images enhance the high resolution to raise more features in the dataset. The images of ROI from four plantar regions were then used for the ANN model training (Figure 2a).

### 2.5. Artificial Neural Network (ANN)

As illustrated in Figure 2b, the ANN model architecture in this study consisted of a one-dimensional input layer, two hidden layers, and one output layer. The input layer was constructed using 10,000 neurons. The number of the output layer was matched with the number of classes that determined the walking speed and the walking duration. First, the plantar region images were converted into one-dimensional value series using the flatten layer. Then, the series was used as the input of the ANN model. The hidden layers were used between the input layer and the output layer to propagate the training mechanism. The artificial neuron had weighted inputs and a procedure to generate neuron output through an activation function [35]. This study used two hidden layers (500 neurons in the first hidden layer and 30 neurons in the second hidden layer). Finally, the output layer was the last layer of neurons, producing a 6-neuron output system to classify the condition of walking speed and walking duration. The dataset consists of 864 plantar pressure images obtained from 12 participants × 3 speeds (0.8, 1.6, and 2.4 m/s) × 2 durations (10 and 20 min) × 3 steps × 4 ROI (T1, M1, M2, and HL). The images recorded from ten participants (i.e., randomly selected) were used to train the model, and two participants were used to verify the performance of the classification model. Specifically, 720 images were recorded from ten participants, while 144 images were collected from two participants to meet the 80:20 testing ratio as suggested by Joo and colleagues [20].

Hyperparameters are variables that determine the structure of the network and organize how the model learns. Properly selected hyperparameter values could assist in solving overfitting and underfitting problems. Therefore, hyperparameter tuning is an essential step in the learning model training process [36]. The parameters in this research model were optimized to maximize the learning model performance by adjusting hyperparameters with the learning rate of: 1 × 10^−3^, batch size: 24, epochs: 1000. Model optimization with Adam was chosen for this study because it was straightforward to implement, computationally efficient, requires little memory, and was well suited for many data and parameters [37]. The model in this study was trained using a personal computer with the operating system Windows 10, 64 bits based on an Intel (R) Core i7-10700k CPU @ 3.80 GHz 3.79 GHz, 32 GB RAM, 10 GB display memory (VRAM), and supported by a super GPU (NVIDIA Ge-Force RTX 3080).

Examples of an accuracy graph and loss graph of the ANN model in this study for walking speed classifications (0.8, 1.6, and 2.4 m/s) of 10 min using a plantar pressure image in the T1 region are presented in Figure 3. The epoch represents a forward pass and a backward pass calculation during training the model. In Figure 3a, the training accuracy is calculated from the correct predictions divided by the total number of predictions made using the training dataset. The validation accuracy is calculated by comparing the model’s predictions to the total number of predictions made using the validation dataset. The ratio between the training dataset with the validation dataset is 80:20. From the graph of model accuracy, the classification model can be trained appropriately because the trend of accuracy in both data sets (training and validation) can reach an accuracy value of 0.94. Furthermore, the training line and the validation line coincide with each other. It shows that neither overfitting nor underfitting occurs in the classification model. Figure 3b illustrates the training loss, i.e., the loss function of the training dataset and model predictions. Conversely, the validation loss is the difference between the model’s predictions and the validation dataset’s loss function. The grey line denotes model validation, and the black line represents model training. The graph of model loss shows that the model has comparable performance on the training and validation datasets with a loss value of 0.05.

### 2.6. Evaluation

The F1-score is a measure of a model’s accuracy in classifying datasets. This measure combines precision and recalls to represent the harmonic mean of the model [38]. The F1-score is usually used to evaluate binary or multiclass classification models on various types of machine learning and deep learning models. The performance of the learning model was evaluated using the F1-score defined by Equation (1), as follows:(1)$$F1-\mathrm{score}=2\times \frac{\mathrm{precision}\times \mathrm{recall}}{\mathrm{precision}+\mathrm{recall}}=\frac{\mathrm{TP}}{\mathrm{TP}+\frac{1}{2}\left(\mathrm{FP}+\mathrm{FN}\right)},$$
where TP is true positive, TN is true negative, FP is false positive, and FN is false negative; precision is defined as the fraction of all positive predictions that are true positives TP/(TP + FP); recall is defined as the fraction of all actual positives that are predicted positive TP/(TP + FN).

## 3. Results

The demographic data were (mean ± standard deviation): age, 27.1 ± 5.8 years; height, 1.703 ± 0.100 m; weight, 63.5 ± 13.5 kg; and body mass index (BMI), 21.7 ± 2.9 kg/m^2^. The results of interviews with all participants stated that all participants were right-handed. Therefore, the leg that dominates all participants is the right side.

The effect of walking speed for 10 min, the T1 region showed a substantial difference amount the different walking speeds between 0.8, 1.6, and 2.4 m/s with the F1-scores of 0.91, 0.92, and 1.00, respectively. On the other hand, the F1-scores in the M1 region between 0.8, 1.6, and 2.4 m/s were 0.36, 0.55, and 0.71, respectively. Thus, the T1 region showed better F1-scores than the M1 regions. However, there was a decrease in the F1-score at the moderate walking speed (1.6 m/s) for 10 min walking duration in the M2 and HL regions (Table 1 and Figure 4).

The effect of walking speed for 20 min, the T1 region showed a substantial difference in walking speeds between 0.8, 1.6, and 2.4 m/s with F1-scores of 0.80, 0.22, and 0.17, respectively. In the M2 region between 0.8, 1.6, and 2.4 m/s, the F1-scores were 0.67, 0.20, and 0.18, respectively. In the HL region between 0.8, 1.6, and 2.4 m/s, the F1-scores were 0.50, 0.29, and 0.22, respectively. The T1 region showed better F1-scores than the M2 and HL regions. There was an increase in the F1-score at the fast walking speed (2.4 m/s) for 20 min walking duration in the M1 region (Table 1 and Figure 4).

The F1-score achieved by the classification model at the four plantar regions is shown in Table 1. These results indexed that the F1-score was higher in the T1 region. The highest results were at 0.8 m/s walking speeds (F1-score = 0.91), 1.6 (F1-score = 0.92), and 2.4 m/s (F1-score = 1.00) at 10 min walking duration.

Regarding the effect of walking durations, the comparison of walking duration between 10 min versus 20 min was presented at walking speeds of 0.8 m/s, 1.6 m/s, and 2.4 m/s, as seen in Figure 5 and Table 2. The results indicate that the walking duration influences the F1-score, especially in all four plantar regions. The F1-score of shorter walking duration (10 min) tends to be higher than the longer duration (20 min).

Table 2 shows the comparison of the F1-scores obtained from all four plantar regions at walking durations of 10 min and 20 min. In the T1 region, there were differences in the F1-score between 10 min and 20 min walking duration in 0.8 m/s walking speed (0.91 vs. 0.80), 1.6 m/s walking speed (0.92 vs. 0.22), and 2.4 m/s walking speed (1.00 vs. 0.17). In the M1 region, there were differences in the F1-score between 10 min and 20 min walking duration in 0.8 m/s walking speed (0.36 vs. 0.50), 1.6 m/s walking speed (0.55 vs. 0.29), and 2.4 m/s walking speed (0.71 vs. 0.40). In the M2 region, there were differences in the F1-score between 10 min and 20 min walking duration in 0.8 m/s walking speed (0.71 vs. 0.67), walking speed of 1.6 m/s (0.22 vs. 0.20), and walking speed of 2.4 m/s (0.80 vs. 0.18). In the HL region, there were differences in the F1-score between 10 min and 20 min walking duration in 0.8 m/s walking speed (0.35 vs. 0.50), 1.6 m/s walking speed (0.00 vs. 0.29), and 2.4 m/s walking speed (0.50 vs. 0.22).

## 4. Discussion

This study showed that higher F1-scores are generated by the classification model when using a fast walking speed (2.4 m/s) compared to moderate and lower walking speeds (1.6 m/s and 0.8 m/s). Fast walking speed (2.4 m/s) significantly increased plantar skin blood flow compared to moderate (1.6 m/s) and slow (0.8 m/s) walking speeds [29]. Hence, the plantar pressure images produced at the fast walking speed are more prominent than those at moderate and slow walking speeds. Previous studies have shown that faster walking speeds significantly increased maximum force and PPP compared to slower walking speeds [39,40,41]. These events affect pixel values in plantar pressure images. Therefore, faster walking speeds can result in higher pixel levels in plantar pressure images than slower speeds [42]. Therefore, the ANN model could more accurately recognize the plantar pressure pattern at a faster walking speed. Furthermore, a striking difference between the F1-scores obtained when walking at a speed of 2.4 m/s for 10 min and those for 20 min was observed. The F1-score for the walking duration of 10 min is higher than those for 20 min. A fast walking speed (2.4 m/s) with a walking duration of 10 min results in a more stable gait, resulting in a more consistent pattern of plantar pressure distribution than walking at 20 min. In addition, the plantar image generated at a fast walking speed with a shorter duration has more visible patterns than those with a longer duration, making it easier for the ANN model to classify each walking speed. Our finding, supported by Bhatt et al. and Young and Dingwell [43,44], suggested that a higher walking speed and a shorter walking duration improve walking stability. Our finding implies that a faster walking speed with a walking duration of 10 min could result in a more consistent pattern in plantar pressure distribution than those with a waking duration of 20 min.

The proposed model achieved the highest F1-score that is occurred in T1 region (0.94), then followed by M2 (0.61), M1 (0.56), and HL (0.41), respectively. Each region of plantar pressure data will generate a different F1-score of the ANN model. The pressure load distribution on distinct regions of the foot varies greatly during walking activities [45]. The highest PPP is in the first toe (T1), then followed by PPP in the forefoot (M1, M2, and HL) [40]. The first toe region was subjected to the heaviest loads, and the highest peak pressure values (over 430 kPa on average) are observed under the first toe pad during the push-off phase [46]. Additionally, Tanaka et al. demonstrated that the first toe (T1) pressure under the foot provides biomechanical information on the foot, such as balance and stable gait for determining human health conditions [47]. The analysis of fast walking speed shows where the maximum peak pressure is absorbed in the first toe region [48]. Chou et al. suggesting that the first toe is vital for balance and can be considered in the future when considering toe amputation or transfer [49].

PPP is one of the in-shoe pressure variables that has been highlighted to be suitable for classifying the development of foot ulcers [5]. Based on this assumption, this study proposed that in-shoe plantar pressure images (i.e., includes detailed PPP in the spatial space) are variables that should be examined in a diabetic patient at risk for foot ulcers. However, this study only referred to the dataset of walking speed and walking duration of healthy participants rather than patients with abnormal gait for training and validation of the classification model. Although this study only refers to the healthy participants, the results showed that the walking speed and walking duration could be classified using the ANN model. Therefore, patients can be warned during their physical activity based on the classification of exercise volume. Furthermore, the results supported by Lung et al. [9], which observed correlation coefficients between healthy and diabetic participants, revealed that the PPP values of the healthy and the diabetic participants were similar. In conclusion, our study can be used by diabetic patients to provide an appropriate assessment of exercise volume.

The boxplots in Figure 6 show the distribution of four F1-scores by 5-fold cross-validation [20,38] when classifying a class label of 10 min speed in every plantar region (i.e., T1, M1, M2, and HL). The outermost horizontal lines indicate the maximum and minimum values of the F1-score. The horizontal lines below and above the box are the lower and upper quartiles, respectively. The center lines are the median values. The closer the box of F1-score is to 1, the higher the accuracy of the model. In addition, the narrower the gap between the maximum and minimum values, the higher the robustness of the classification model. For this study comparing the boxplots of F1-scores for each plantar region, the dataset of T1 has higher accuracy with a mean F1-score of 0.94 and a narrower box than three other datasets (i.e., M1, M2, and HL). In addition, the dataset of M1 and M2 achieve similar performance.

Meanwhile, the dataset of HL generates a lower performance compared to the other three datasets. The T1 region is more dominant in walking or running movements, especially during the pre-swing phase but has a thinner tissue thickness [9,50]. Repeated pressure on these soft tissues causes the T1 region to have a greater potential for foot ulcers. Visually, the global data used in this study are often at the boundary of the criteria for classifying foot plantar images, which is somewhat ambiguous in features, resulting in poor performance compared to if the image is analyzed regionally. Therefore, a deep analysis through regional feature extraction is needed to ensure that the plantar patterns can be extracted consistently. The main finding of this study is that the dataset on the T1 region achieves the best performance with the highest accuracy and minor variance. Thus, it can be seen that analysis of plantar region pressure images is essential to increase the classification performance of walking speed and walking duration.

This study was compared with other related studies to evaluate the performance of the classification model (Table 3). However, they used the ANN model with a different framework to analyze problems related to plantar pressure data, as Joo et al. used an ANN model with one hidden layer to analyze gait speed [20]. Meanwhile, Begg and Kamruzzaman used a neural network to analyze gait changes [51]. The proposed ANN model uses two hidden layers architecture to classify walking speed and walking duration. It has been proven that the proposed ANN model increases the performance of the classification model based on the plantar region pressure image. Compared with other studies, Joo et al. analyzed plantar pressure data using the ANN model with one hidden layer [20]; this model achieved an accuracy of 0.71. In comparison, Begg and Kamruzzaman using an ANN model with three hidden layers, obtained a better model accuracy of 0.83 [51].

This study demonstrated better performance of the classification model than other studies, with an F1-score of 0.94. These results are still better than the preliminary study Chen et al. [52]. The preliminary study used a convolutional neural network (CNN) to analyze plantar pressure images. Although CNN has a better ability to extract features in two-dimensional data, the inconsistency of the dataset prevents this method from producing high accuracy. One of the major challenges in recording plantar pressure data is the data generation which is sometimes inconsistent due to slips between the sensor layer and the insole, variations in movement that are not uniform between participants, and the physical condition of the participants [53]. The ANN model, which focuses on extracting plantar data regionally, is proven to be superior in producing higher F1-scores, especially in datasets that are generally inconsistent. In addition, training local patterns comprehensively using ANN can improve the accuracy of the classification system. The results demonstrate the competitiveness of this study with previous studies, especially for the classification of walking speed and walking duration.

This research has some limitations. First, this study only refers to the dataset of walking speed and walking duration of healthy participants rather than those of patients with abnormal gait for training and validation of the classification model. Second, there are more data features in time series plantar pressure images due to different walking intensities over time. The ANN model could not capture temporal patterns inside these spatiotemporal plantar pressure datasets. However, it is possible that these temporal patterns can provide additional information that can be used to improve the performance of the classification model. A spatiotemporal classification model can be used to study sequential data because it can extract time-series patterns in spatiotemporal datasets, such as plantar pressure images. These spatiotemporal datasets could be analyzed accurately by using the deep spatiotemporal classification model [22] and spatiotemporal data analysis [54]. Third, the plantar pressure image generated using the insole-type F-scan system measurement device is not very detailed in terms of resolution and measurement in this study. High image resolution can reveal more detailed image features that can improve the performance of neural networks in various machine learning tasks [55]. Therefore, a general clinical model based on actual patient data needs to be developed in future studies. It is expected that the method proposed in this study would be the foundation for the establishment of new research methods for the assessment of the development of foot ulcers, especially in DM patients.

## 5. Conclusions

This study revealed that the proposed ANN-based method can classify walking speed and walking duration using plantar pressure images. Furthermore, the model achieved the highest F1-score using fast walking speed compared to moderate and slow walking speed. Visually, the faster walking speed with a walking duration of 10 min resulted in a more consistent plantar pressure distribution pattern than a walking duration of 20 min. Therefore, this consistent information produces a higher performance achieved by the proposed model. Specifically, the plantar pressure image at the first toe region (T1) provided a more detailed pressure pattern at each walking speed; therefore, the proposed model could easily recognize the images. Finally, this results of this study could be used as a detection system suitable for walking exercise and rehabilitation interventions for patients at risk for foot ulcers.

## Figures and Tables

**Figure 1 sensors-21-06513-f001:**
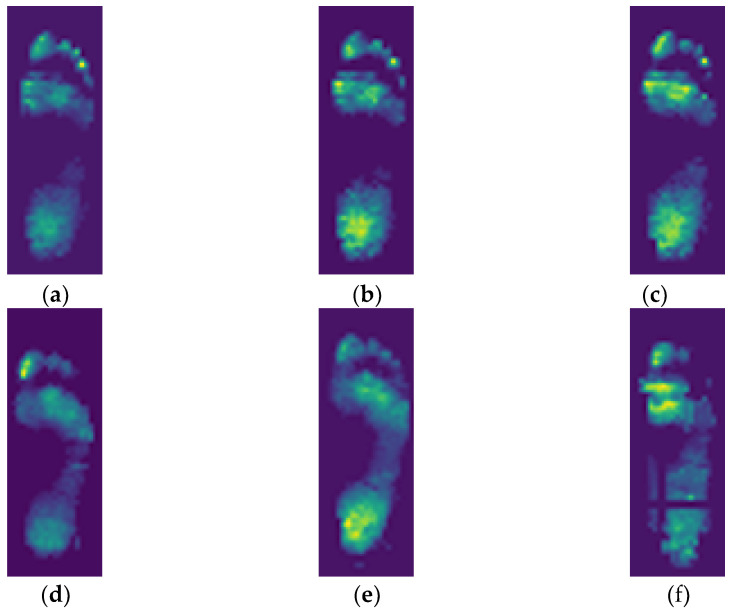
Examples of plantar pressure images after different walking speeds and durations in a representative participant. (**a**) 0.8 m/s for 10 min; (**b**) 1.6 m/s for 10 min; (**c**) 2.4 m/s for 10 min, (**d**) 0.8 m/s for 20 min; (**e**) 1.6 m/s for 20 min; (**f**) 2.4 m/s for 20 min.

**Figure 2 sensors-21-06513-f002:**
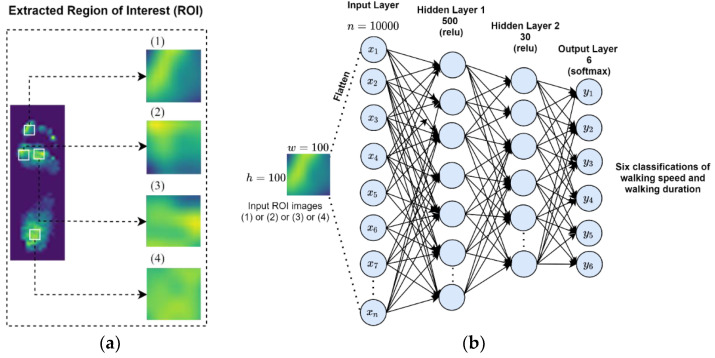
Region of Interest (ROI) input images and ANN model architecture. (**a**) ROI images: the input contributes to the classification model. The extracted ROI consists of four plantar pressure regions defined as: (1) T1, (2) M1, (3) M2, and (4) HL; (**b**) the ANN model architecture is utilized for classifying walking speeds and walking durations. T1, first toe; M1, first metatarsal head; M2, second metatarsal head; HL, heel.

**Figure 3 sensors-21-06513-f003:**
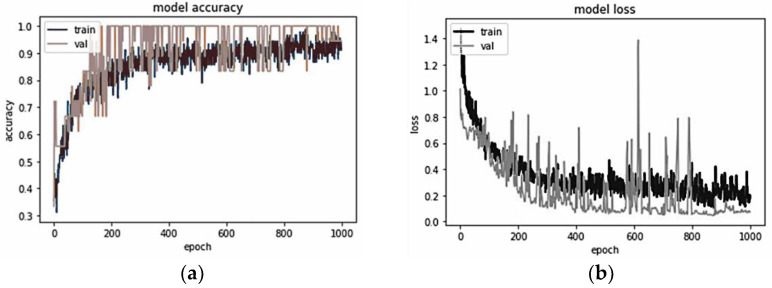
The performance of the proposed ANN model during training and validation using the T1 dataset. (**a**) graph of model accuracy is the number of classifications that is successfully predicted, divided by the total number of predictions; (**b**) graph of model loss indicates how bad the model’s prediction is. Train, the model training indexed by the black line; Val, the model validation indexed by the grey line; T1, first toe.

**Figure 4 sensors-21-06513-f004:**
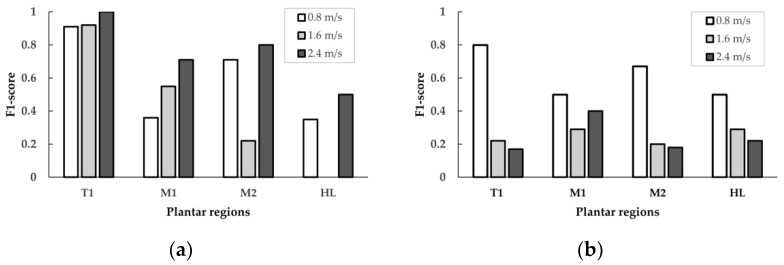
Comparison of the F1-scores of walking speed (0.8, 1.6, and 2.4 m/s) in four plantar regions for two walking durations. (**a**) 10 min walking duration; (**b**) 20 min walking duration. T1, first toe; M1, first metatarsal head; M2, second metatarsal head; HL, heel.

**Figure 5 sensors-21-06513-f005:**
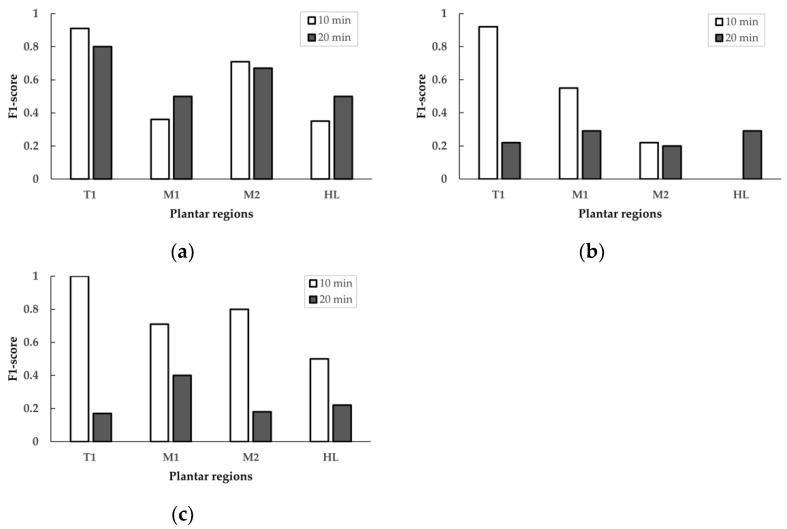
Comparison of F1-scores on walking duration (10 min and 20 min) in the four plantar regions at walking speed (**a**) 0.8 m/s, (**b**) 1.6 m/s, and (**c**) 2.4 m/s. T1, first toe; M1, first metatarsal head; M2, second metatarsal head; HL, heel.

**Figure 6 sensors-21-06513-f006:**
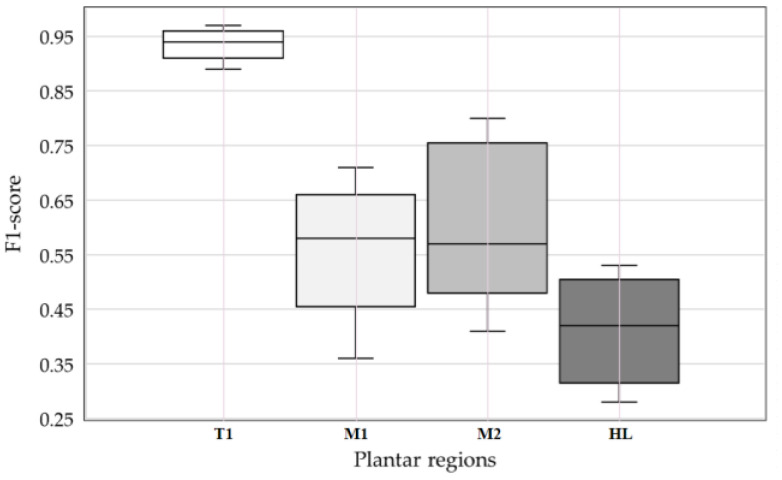
Result of the ANN model F-score in 5-fold cross-validation.

**Table 1 sensors-21-06513-t001:** Effect of walking speed on the F1-score in four plantar regions.

Walking Duration	Plantar Region	F1-Score of Walking Speed		
		0.8 m/s	1.6 m/s	2.4 m/s
10 min	T1	0.91	0.92	1.00
	M1	0.36	0.55	0.71
	M2	0.71	0.22	0.80
	HL	0.35	0.00	0.50
20 min	T1	0.80	0.22	0.17
	M1	0.50	0.29	0.40
	M2	0.67	0.20	0.18
	HL	0.50	0.29	0.22

Note: T1, first toe; M1, first metatarsal head; M2, second metatarsal head; HL, heel.

**Table 2 sensors-21-06513-t002:** Effect of walking duration on the F1-score in four plantar regions.

Walking Speed	Plantar Region	F1-Score of Walking Duration	
		10 min	20 min
0.8 m/s	T1	0.91	0.80
	M1	0.36	0.50
	M2	0.71	0.67
	HL	0.35	0.50
1.6 m/s	T1	0.92	0.22
	M1	0.55	0.29
	M2	0.22	0.20
	HL	0.00	0.29
2.4 m/s	T1	1.00	0.17
	M1	0.71	0.40
	M2	0.80	0.18
	HL	0.50	0.22

Note: T1, first toe; M1, first metatarsal head; M2, second metatarsal head; HL, heel.

**Table 3 sensors-21-06513-t003:** Comparison of the performance of the classification model based on plantar pressure images with relevant studies.

References	Study Case	Target	Method	Accuracy	F1-Score
Begg and Kamruzzaman (2006) [51]	Plantar foot data	Gait changes	ANN (3 hidden layers)	0.83	-
Joo et al. (2014) [20] 99 sensing nodes	Plantar pressure data	Gait speed	ANN (1 hidden layer)	0.71	-
Chen et al. (2021) [52] A preliminary study global pattern (660 sensing nodes)	Plantar pressure images	Walking speed	Convolutional Neural Network	-	0.86
This study region pattern (660 sensing nodes)	Plantar region pressure images	Walking speed and walking duration	ANN (2 hidden layers)	0.94	0.94

## Data Availability

The data presented in this study are available on request from the corresponding author.

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
