# Peer review of "Estimation of Various Walking Intensities Based on Wearable Plantar Pressure Sensors Using Artificial Neural Networks"

_sensors, 2021, doi:10.3390/s21196513_

Round 1

Reviewer 1 Report

The paper proposes a machine learning model to classify the walking speed and duration using plantar region pressure images. And this paper confirms that the proposed ANN-based method can classify walking speed and walking duration using plantar pressure images. The work is practical, although I have some comments below.

1) Please check the entire paper for grammatical errors. Please use one form of tense (past tense or present tense). For instance L305 "This study showed..." L322 "This study shows...".

2) Given the size of the training set, cross-validation is needed.

3)Fig. 5 is barely readable.

4) The sentence in L312 needs references.

Author Response

Reviewer 1

The paper proposes a machine learning model to classify the walking speed and duration using plantar region pressure images. And this paper confirms that the proposed ANN-based method can classify walking speed and walking duration using plantar pressure images. The work is practical, although I have some comments below.

[1]. One form of tense

1) Please check the entire paper for grammatical errors. Please use one form of tense (past tense or present tense). For instance L305 "This study showed..." L322 "This study shows...".

## Authors’ Response:

Thank you very much for the reviewers' suggestions. We have checked the entire paper for grammatical errors and have updated every sentence, consistently.

[2]. Size

2) Given the size of the training set, cross-validation is needed.

## Authors’ Response:

Thank you very much for the reviewer's suggestions. We have added cross-validation in Section Discussion, fourth paragraph (LINE 352-376).

The boxplots in Figure 6 show the distribution of four F1-scores by 5-fold cross-validation [20, 38] when classifying a class label of 10-min speed in every plantar region (i.e., T1, M1, M2, and HL). The outermost horizontal lines indicate the maximum and minimum values of the F1-score. The horizontal lines below and above the box are the lower and upper quartiles, respectively. The center lines are the median values. The closer the box of F1-score is to 1, the higher the accuracy of the model. In addition, the narrower the gap between the maximum and minimum values, the higher the robustness of the classification model. For this study comparing the boxplots of F1-scores for each plantar region, the dataset of T1 has higher accuracy with a mean F1-score of 0.94 and a narrower box than three other datasets (i.e., M1, M2, and HL). In addition, the dataset of M1 and M2 achieve similar performance.

Meanwhile, the dataset of HL generates a lower performance compared to the other three datasets. The T1 region is more dominant in walking or running movements, especially during the pre-swing phase but has a thinner tissue thickness [9, 50]. Repeated pressure on these soft tissues causes the T1 region to have a greater potential for foot ulcers. Visually, the global data used in this study are often at the boundary of the criteria for classifying foot plantar images, which is somewhat ambiguous in features, resulting in poor performance compared to if the image is analyzed regionally. Therefore, a deep analysis through regional feature extraction is needed to ensure that the plantar patterns can be extracted consistently. The main finding of this study is that the dataset on the T1 region achieves the best performance with the highest accuracy and minor variance. Thus, it can be seen that analysis of plantar region pressure images is essential to increase the classification performance of walking speed and walking duration.

Figure 6. Result of ANN model F- score in 5-folding cross-validation.

[3]. Barely readable

3) Fig. 5 is barely readable.

## Authors’ Response:

Thanks for the reviewer's comments. The Fig. 5 has improved, so it's readable.

[4]. Needs references

4) The sentence in L312 needs references.

## Authors’ Response:

Thank you very much for reviewer’s suggestion, the references have been added in LINE 308-310 to strengthen the statement.

Previous studies have shown that faster walking speeds significantly increased maximum force and PPP compared to slower walking speeds [39-41].”

Reviewer 2 Report

This study aimed to examine different walking speeds and duration on plantar region pressure images using the ANN model for detecting appropriate exercise volume in patients at risk for foot ulcers. However, the overall logic of the article is not good, and the correlation between the gait characteristics of the diabetic population and the results of this study needs to be strengthened.

Major comments:

  1. The author should first clarify the logical relationship and evidence between plantar pressure and diabetic foot ulcers in order to quantify the walking intensity for rehabilitation interventions to indicate suitable walking. Secondly, this is methodology research of quantifying the gait load, so it is recommended author organize the logic of the background introduction based on the research purposes.
  2. As the impact of plantar soft tissue stiffness on the foot biomechanics, the relationship between the plantar mechanical stress (Patry et al., 2013) and the changes in plantar tissue stiffness (Lung et al., 2020) should be evaluated in order to better quantify the appropriate exercise volume in patients at risk for foot ulcers. Are the results of this study effective and comprehensive for assessing the walking intensity of people with diabetes?

Minor comments:

  1. Can the gait characteristics of the participants in this study represent that of diabetic patients?
  2. What is the basis and purpose of the selection of walking speed and duration?
  3. What are the abnormal gait characteristics of diabetic feet? Can the abnormal pressure in the T1 be used to assess the risk of foot ulcers in diabetic patients?

Author Response

Reviewer 2

Correlation & strengthened

This study aimed to examine different walking speeds and duration on plantar region pressure images using the ANN model for detecting appropriate exercise volume in patients at risk for foot ulcers. However, the overall logic of the article is not good, and the correlation between the gait characteristics of the diabetic population and the results of this study needs to be strengthened.

Major comments:

[1]. Logical relationship and evidence

  1. The author should first clarify the logical relationship and evidence between plantar pressure and diabetic foot ulcers in order to quantify the walking intensity for rehabilitation interventions to indicate suitable walking. Secondly, this is methodology research of quantifying the gait load, so it is recommended author organize the logic of the background introduction based on the research purposes.

## Authors’ Response:

-- Thank you very much for your fruitful comments. The authors have already revised the Introduction section by providing logical relationships in this study and add more evidence. The revised manuscript can be seen in LINE 55-70 in Section Introduction.

“Measurement of walking intensity is a major challenge fundamentally because the development of foot ulcers is influenced by repetitive loads on the plantar soft tissues.

Researchers have attempted to address this situation through analysis of plantar pressure patterns to study the development of foot ulcers [10, 11]. The plantar pressure patterns are essential information to detect the development of foot ulcers, which have enabled to reach state-of-the-art predictive performance in classification tasks [12]. Furthermore, plantar pressure values, such as PPP, vary during gait in different areas of the plantar foot [13]. Research studies analyzed PPP values and demonstrated that four plantar regions are at a higher risk of foot ulcers, namely first toe (T1), first metatarsal head (M1), second metatarsal head (M2), and heel (HL) [14, 15]. Fundamentally, the tissue thickness of the T1 is thinner than M1, M2, and HL [9]. These regions could thus play a significant role in the development of foot ulcers. Analyzing the four plantar regions may provide more detailed patterns of the foot plantar thus could be used to assess the appropriate intensity of exercise for patients at risk for foot ulcers.”

-- Thank you very much for your constructing comments. The authors have already revised the manuscript by organizing the research purposes based on reviewer’s advice in LINE 93-103 Section Introduction, last paragraph.

“Based on the above studies, walking exercise is an excellent intervention to improve the health of people with DM, but the appropriate walking intensity is critical. Therefore, it is necessary to classify walking speed and walking duration to find the proper walking intensity. In this work, the credential of ANN is implemented as a potential classifier of walking speed and walking duration in three different ways: 1) Demonstrate that ANN can accurately predict walking speed and walking duration using plantar region pressure images and compare predictions with preliminary approaches; 2) Establish the region analysis of predictions using a dataset of four different regions; 3) Assess the proposed scheme’s reliability using cross-validation of F1-scores. The results of this study provide a foundation for understanding the pattern of plantar pressure during various walking activities for detecting appropriate exercise volume in patients at risk for foot ulcers.”

[2]. Plantar tissue stiffness

  1. As the impact of plantar soft tissue stiffness on the foot biomechanics, the relationship between the plantar mechanical stress (Patry et al., 2013) and the changes in plantar tissue stiffness (Lung et al., 2020) should be evaluated in order to better quantify the appropriate exercise volume in patients at risk for foot ulcers. Are the results of this study effective and comprehensive for assessing the walking intensity of people with diabetes?

## Authors’ Response:

Thank you very much for the reviewer's comments. The authors have revised the manuscript in LINE 338-351 to answer the reviewer’s questions.

“PPP is one of the in-shoe pressure variables that has been highlighted to be suitable for classifying the development of foot ulcers [5]. Based on this assumption, this study proposed that in-shoe plantar pressure images (i.e., includes detailed PPP in the spatial space) are variables that should be examined in a diabetic patient at risk for foot ulcers. However, this study only referred to the dataset of walking speed and walking duration of healthy participants rather than patients with abnormal gait for training and validation of the classification model. Although only refers to the healthy participants, the results showed that the walking speed and walking duration could be classified using the ANN model. Therefore, patients can be warned during their physical activity based on the classification of exercise volume. Furthermore, the results supported by Lung et al. [9], which observed correlation coefficients between healthy and diabetic participants, revealed that the PPP values of the healthy and the diabetic participants were similar. In conclusion, our study can be used by diabetic patients to provide an appropriate assessment of exercise volume.”

Minor comments:

[1]. Diabetic patients

  1. Can the gait characteristics of the participants in this study represent that of diabetic patients?

## Authors’ Response:

Thank you for the reviewer comments. The proposed model can be used as a basic understanding to monitor foot plantar pressure using plantar region pressure image. Based on the previous studies [7], walking exercise is an excellent intervention to improve the health of people with DM, but the appropriate walking intensity is critical. The results showed that the walking speed factor caused a significant main effect of plantar stiffness, even though the study only recruited healthy participants. In addition, another study observed correlation coefficients between healthy and diabetic subjects [9]. The results revealed that the PPP values of the healthy and the diabetic subjects were similar. Finally, the results of our study provide a foundation for understanding the pattern of plantar pressure during various walking activities for detecting appropriate exercise volume in patients at risk for foot ulcers.

[2]. walking speed and duration

  1. What is the basis and purpose of the selection of walking speed and duration?

## Authors’ Response:

Thank you very much for the review question. These classifications (different speeds and durations) have been universally recommended as an exercise to improve physical and psychological health in people with diabetes mellitus (DM) and peripheral arterial disease. We have mentioned it in LINE 131-135, as seen below.

The experimental procedures were based on our previous study [9, 27]. Meanwhile, this study examined three walking speeds (slow at 0.8 m/s, moderate at 1.6 m/s, and fast at 2.4 m/s) and two walking durations (10 and 20 min). Three speeds were selected to simulate three common speeds in adults [28], and two durations were selected based on average walking time and physical activity [29].”

[3]. abnormal pressure

  1. What are the abnormal gait characteristics of diabetic feet? Can the abnormal pressure in the T1 be used to assess the risk of foot ulcers in diabetic patients?

## Authors’ Response:

Thank you very much for the review question. T1 is one of the four regions that have a high risk of the development of foot ulcers. Therefore, the T1 pressure region can be used to assess ulcer risk in diabetic patients. The results of the previous study demonstrated that the walking speeds (slow at 0.8 m/s, moderate at 1.6 m/s, and fast at 2.4 m/s) significantly affected plantar tissue stiffness. The plantar tissue stiffness after walking at 1.6 m/s for 20 min was significantly lower compared to walking at 0.8 m/s for 20 min. This finding is significant because moderate-to-fast walking speed (1.6 m/s) can decrease plantar tissue stiffness compared to slow walking speed (0.8 m/s).

Meanwhile, slow walking speed can increase plantar tissue stiffness that may cause a higher risk of foot ulcers. This study suggests people at risk for foot ulcers walking at a preferred or fast speed (1.6 m/s) rather than walking slowly (0.8 m/s). On the other hand, our study classifies foot plantar pressure in several classes of walking speed and walking duration. The results of this study are more in analyzing the influence of each region that has a high chance of developing foot ulcers during walking activity. This paper is a preliminary study to assess the exercises that are suitable for DM patients.

Reviewer 3 Report

General comments:

This is a very interesting study. This area of research is important for the future of rehabilitation in patients with foot ulcers and peripheral neuropathy. The manuscript is well written and I only have a few specific comments for the authors to address.

Specific comments:

Methods:

  1. Change all units of speed to SI base unit i.e. mph to m/s. The major of the world uses SI base units not imperial units. Therefore, it’s best to use metric and if you wish, add the imperial units in brackets.
  2. The authors are using subject and participant within the manuscript. Best to be consistent and pick one, suggest to only use participant.

Results:

  1. Line 242 – ‘weight’ is used incorrectly. Weight is a force (N). body mass is the correct term to use. Please update.
  2. Line 242- How was limb dominance determined?

Author Response

Reviewer 3

General comments:

This is a very interesting study. This area of research is important for the future of rehabilitation in patients with foot ulcers and peripheral neuropathy. The manuscript is well written and I only have a few specific comments for the authors to address.

Specific comments:

[1]. Methods: unit

  1. Change all units of speed to SI base unit i.e. mph to m/s. The major of the world uses SI base units not imperial units. Therefore, it’s best to use metric and if you wish, add the imperial units in brackets.

## Authors’ Response:

Thanks for the reviewer's suggestions. The units have converted mph to m/s.

1.8 mph = 0.8 m/s

3.6 mph = 1.6 m/s

5.4 mph = 2.4 m/s

[2]. Methods: participant

  1. The authors are using subject and participant within the manuscript. Best to be consistent and pick one, suggest to only use participant.

## Authors’ Response:

Thanks for the reviewer's suggestions. The manuscript has been updated using the consistent term (i.e., “participants”).

[1]. Results: body mass

  1. LINE 242 – ‘weight’ is used incorrectly. Weight is a force (N). body mass is the correct term to use. Please update.

## Authors’ Response:

Thank you very much for the reviewer's correction. Line 251 has updated the term using body mass.

[2]. Results: limb dominance

  1. Line 242- How was limb dominance determined?

## Authors’ Response:

Thank you for the reviewer's question about how was limb dominance determined? In this study, limb dominance was determined and adding as LINE 251-253:

“The results of interviews with all participants stated that all participants were right-handed. Therefore, the leg that dominates all participants is the right side.”

Author Response

Reviewer 4

Glad to have an opportunity to review this manuscript. There are several methodological concerns that limit the readers understanding of why this research was conducted.

Below I have provided comments for the authors.

From my point of view, this study can be improved for publication, however to be published on this quality journal the following issues needs to be properly modify.

[1]. Achievement on clinical situations

The abstract is successfully compiling and summarizing focal points of this study Validity issue your study is successful to provide validity evidence of your measurement issues. I am convinced to your findings are clear about what would be potential lessons from reading your study. But, could you explain the benefits of your achievement on clinical situations?

## Authors’ Response:

Thank you very much for your thoughtful review of our manuscript. The benefits of this study could be used as a detection system suitable for walking exercise and rehabilitation interventions for patients at risk for foot ulcers. A previous study revealed a lower stiffness of plantar tissue after walking at 1.6 m/s (moderate speed) compared to 0.8 m/s (slow speed) because of a higher increase in plantar skin blood flow during walking at 1.6 m/s. On the other hand, our study makes classifications of the exercises that are suitable for DM patients based on the foot plantar pressure images. In a more detailed result, this study provides classifications based on the four regions that have a higher risk of ulceration. The main finding of this study is that the dataset on the T1 region achieves the best performance with the highest accuracy and a minor variance. The T1 region is more dominantly used in walking or running movements, especially during the pre-swing phase but has a thinner tissue thickness.

[2]. Influence of plantar pressures

Introduction section may be improved adding new information in order to provide an adequate state- of-the-art including some references about the influence of plantar pressures (doi: 10.3390/healthcare9080932).

## Authors’ Response:

Thank you very much for the reviewer's suggestion. We have added new information to provide an adequate state-of-the-art, including some references about the influence of plantar pressure at LINE 58-70 in the introduction section paragraph 2 as described below.

“Researchers have attempted to address this situation through analysis of plantar pressure patterns to study the development of foot ulcers [10, 11]. The plantar pressure patterns are essential information to detect the development of foot ulcers, which have enabled to reach state-of-the-art predictive performance in classification tasks [12]. Furthermore, plantar pressure values, such as PPP, vary during gait in different areas of the plantar foot [13]. Research studies analyzed PPP values and demonstrated that four plantar regions are at a higher risk of foot ulcers, namely first toe (T1), first metatarsal head (M1), second metatarsal head (M2), and heel (HL) [14, 15]. Fundamentally, the tissue thickness of the T1 is thinner than M1, M2, and HL [9]. These regions could thus play a significant role in the development of foot ulcers. Analyzing the four plan-tar regions may provide more detailed patterns of the foot plantar thus could be used to assess the appropriate intensity of exercise for patients at risk for foot ulcers.”

[3]. Main finding of the study

There are not a good description of the properties of the outcome measurements, In fact authors should describe a detailed statistical analyses Furthermore, Tables, figures and redaction of the results needs show the main finding of the study. adding ICC between pre and post measurement.

## Authors’ Response:

-- Thank you very much for the reviewer's comments. The authors have improved the system reliability using cross-validation as described in a paragraph in Section Discussion, LINE 351-372. As stated in Section Discussion LINE 372-375, the resulting outcome answers the hypotheses.

“The main finding of this study is that the dataset on the T1 region achieves the best performance with the highest accuracy and the smallest variance. It can be seen that analysis of plantar region pressure images is essential to increase the classification performance of walking speed and walking duration.”

-- For the request of the study's main finding, LINE 401-403 shows our contributions in this field.

“The results demonstrate the competitiveness of this study with previous studies, especially for the classification of walking speed and walking duration.”

-- For the request of adding ICC between pre and post-measurement, the experimental procedures were based on our previous study, and the pre-measurement didn’t conduct analysis.

[4]. Hypothesis

Please provide a hypothesis.

## Authors’ Response:

Thank you very much for the reviewer's suggestion. The authors provide hypotheses in Section Introduction, LINE 66-70 as presented below.

“This argument led this study to introduce the hypothesis: the T1 region may play a significant role in the development of foot ulcers. Hence, analyzing the four plantar regions may provide more detailed patterns of the foot plantar thus could be used to assess the appropriate intensity of exercise for patients at risk for foot ulcers.

[5]. Future research

Discussion section may include future research studies secondary to the current findings of this study. Clinical considerations, limitations and overall discussion are well-presented, but future research may be useful in order to propose future research regarding this field.

## Authors’ Response:

Thanks for the reviewer's comments. We have added future studies on LINE 404-422, which can be seen as follows:

“This research has some limitations. First, this study only refers to the dataset of walking speed and walking duration of healthy participants rather than those of patients with abnormal gait for training and validation of the classification model. Second, there are more data features in time series plantar pressure images due to different walking intensities over time. The ANN model could not capture temporal patterns inside these spatiotemporal plantar pressure datasets. However, it is possible that these temporal patterns can provide additional information that can be used to improve the performance of the classification model. A spatiotemporal classification model can be used to study sequential data because it can extract time-series patterns in spatiotemporal datasets, such as plantar pressure images. These spatiotemporal datasets could be analyzed accurately by using the deep spatiotemporal classification model [22] and spatiotemporal data analysis [54]. Third, the plantar pressure image generated using the insole-type F-scan system measurement device is not very detailed in terms of resolution and measurement in this study. High image resolution can reveal more detailed image features that can improve the performance of neural networks in various machine learning tasks [55]. Therefore, a general clinical model based on actual patient data needs to be developed in future studies. It is expected that the method proposed in this study would be the foundation for the establishment of new research methods for the assessment of the development of foot ulcers, especially in DM patients.”

[6]. Gait speed

However I suggest to authors should include some reference related to prior studies, As the case of parkinson's disease and gait speed I suggest to include the following reference. (DOI: 10.3390/brainsci10020069)

## Authors’ Response:

Thank you very much for the reviewer's suggestions. We have added references related to previous studies at LINE 43-45.

“Walking has been universally recommended as a rehabilitation strategy to improve physical and psychological health in people with Parkinson’s disease [1], diabetes mellitus (DM), and peripheral arterial disease [2, 3].”

[7]. "research process" and conclusion part

Please check and re-confirm and have other experienced scholars to read your manuscript prior to "submission" in terms of "research process" and conclusion part.

## Authors’ Response:

Thank you very much for the reviewer's suggestions. All authors have examined the manuscript, and experienced scholars have read the manuscript.

Round 2

Reviewer 2 Report

I am satisfied with the revisions. 

Reviewer 4 Report

Authors have adressed all my requeriment in the correct way